# An Observational Study on Oral Health and Quality of Life for RPD Wearers in the N-E Region of Romania

**DOI:** 10.3390/medicina58091247

**Published:** 2022-09-09

**Authors:** Florinel Cosmin Bida, Doriana Agop-Forna, Bogdan Petru Bulancea, Carina Balcoș, Norina Consuela Forna

**Affiliations:** 1Department of Implantology, Removable Prostheses, Dental Prostheses Technology, Faculty of Dental Medicine, “Grigore T. Popa” University of Medicine and Pharmacy, 700115 Iasi, Romania; 2Department Surgery, Faculty of Dental Medicine, “Grigore T. Popa” University of Medicine and Pharmacy, 700115 Iasi, Romania

**Keywords:** quality of life, removable partial dentures (RPD), OHRQoL

## Abstract

There is a lack of information regarding the oral health of the elderly population in Romania; only a few articles have been published about their edentulism, and there are no official data regarding the oral health, OHRQoL, and prosthodontic status of this population. The objective of this study was to assess the relationship between the essential functional qualities of removable partial dentures (RPD) and participants’ oral-health-related wellbeing among an edentulous population from the N-E region in Romania using the OHIP-5-questionnaire. In total, 546 patients from Gr. T. Popa Iasi University were enrolled after following a set of sorting procedures. We used the Kruskal–Wallis test and the Mann–Whitney test to determine whether there were significant differences between the various groups of patients. For questions 1, 2, 3, and 5 of the OHIP, the null hypothesis was rejected, as there were statistically significant differences between the three groups. This study was limited to a specific group. However, it can provide an insight into RPD patients’ happiness when treated in an academic environment.

## 1. Introduction

The World Health Organization has defined health as being a “complete state of physical, mental, and social well-being, and not merely the absence of disease or infirmity.” This concept includes a wide range of stages, from wellbeing to illness. However, there are no global tools for measuring health. Mortality trends, life expectancy, and morbidity indices are often used to describe a nation’s health. Quality of life assessment techniques have been developed to answer the demand for more complete health assessments [1].

According to the World Health Organization Quality of Life Assessment Group, “quality of life (QoL) is defined as the perspective of persons about their place in life, within the cultural context and value systems where they live, and as a function of their goals, expectations, standards and worries” [2]. The loss of teeth can have an impact on regular functional activities, as well as on the meals that can be chosen and enjoyed during mealtimes, according to several published studies [3].

Oral health is an essential component of overall health; however, the standard methods for evaluating an individual’s quality of life in relation to their health do not consider the specific impact that oral health issues have on an individual’s quality of life. Therefore, additional methods are required [4]. Even though oral-health-related quality of life (OHRQoL) does not reflect actual oral health status on its own, it provides the patient’s perspective of their own oral health as well as the significance and influence it has on their lives. OHRQoL may serve as a public health indicator, indicating both the limits of oral health in communities and the influence of oral health and dental therapies on people’s lives [5].

The medical field’s primary focus has shifted in recent years from diagnosis and treatment to other aspects of patient care. The patient’s quality of life is a primary focus of attention. More techniques are becoming available to assess oral-health-related quality of life (OHRQoL) during routine dental procedures using the fewest possible items. The oral-health-related quality of life assessment instrument was created by experts and academics based on health-related quality of life ideas. The Oral Health Impact Profile (OHIP), established by Australian academics Slade and Spencer, is the most often used method for predicting oral-health-related quality of life in the domestic and international related literature [6]. One such measure is known as the Oral Health Impact Profile (OHIP-5), which consists of five questions [6,7]. 

The assessment of the respondents’ subjective feelings provides an estimate of their quality of life. The respondents’ subjective feelings about a disease and its therapy have a greater impact on their quality of life than numerous clinical indices. The interviewees’ satisfaction with oral healthcare services and the score of the oral-health-related quality of life are regarded as two different indices, which represent the efficacy of services in the older population [8]. According to research, dentists’ evaluations of dentures are not necessarily associated with patient contentment [9,10]. There must be consideration given to the difference between patient perception and dental evaluations. So far, research has been conducted regarding many topics concerning the quality of life associated with oral health. Some researchers have focused on understanding the concept of quality of life associated with oral health [11]. Others have made associations between oral health and quality of life, as measured by generic health tools [12,13]. 

We do not have enough information regarding the oral health in the elderly population in Romania, as there are only a few articles published about edentulism in this population, and there are no official data regarding the oral health, OHRQoL, and prosthodontics’ status of the population.

The objective of this study was to assess the relationship between the essential functional qualities of RPDs and participants’ oral-health-related wellbeing among an edentulous population from the N-E region of Romania using the OHIP-5-questionnaire, taking into consideration whether the algorithm for RPDs developed by the Faculty of Dentistry was followed. We began with the hypothesis that there are statistically significant differences in the quality of life between the three groups.

## 2. Materials and Methods

### 2.1. Study Design

This observational study was conducted using a sample of 546 patients who received removable partial dentures (RPDs) in the Department of Prosthodontics, at the Faculty of Dentistry in Iasi, Romania, between January 2004 and January 2019. Undergraduate students and interns worked together under the direction of teaching personnel to provide medical care to the patients. This research study was authorized by the “Grigore T.Popa” Iasi University’s Ethical Committee.

To provide an RPD that is both functional and comfortable, rigorous assessment, design, and care are required. Steps involved in RPD-related therapy include assessment of the abutment teeth, positioning and preparation of the abutments, adjustment of the RPD metal framework, connecting the edentulous areas to the metal framework, interaction with the laboratory, health education for home care and maintenance, and regular preventive recall. Patients who are partially dentate may have lost teeth because of improper oral hygiene; therefore, it is essential for them to practice good home maintenance hygiene, caries intervention strategies, and appropriate use of their removable prostheses to minimize the risk of developing future complications [14,15].

It is essential to the operation’s effectiveness to perform careful individualized planning and manufacture of the RPD for each patient. The RPD design that may best satisfy the demands of an individual patient should be determined by factors such as the architecture of hard and soft tissues, occlusal relationships, tooth location, and the patient’s goals for aesthetics and comfort [16].

### 2.2. Participant Selection

In the beginning, 546 subjects (304 men and 242 women) were selected after following a set of sorting procedures: we recorded data sheets containing all information regarding the treatment and the protocol used and specific laboratory data sheets containing all features of the RPDs as well as the design and distribution of elements for support and stability. RPDs with metal frameworks were fabricated following therapeutic treatment recommendations, with acrylic dentures as an interim treatment followed 1 year later by metal-framework RPDs with clasps or special elements, including hinge, ball and socket, interlocks, bar attachment, and the university prosthetic algorithm for RPDs.

When selecting patients in the second phase, each participant was contacted and asked to proceed by completing an OHIP-5 questionnaire; they were also asked for their written consent. After excluding those who did not respond and those who were treated only with partial acrylic base dentures, 338 (61.90%) valid participants were included in the final analysis. They were then divided into three categories:
The first group included 106 patients (following the RPD treatment algorithm) who agreed to the treatment plan and received acrylic dentures first and then an RPD with metal framework and special elements. This group was considered the control group, due to the RPD’s strong stability and retention and patients’ compliance with the clinic’s protocol.The second group included 181 patients (following the RPD treatment algorithm) who agreed to the treatment plan and received acrylic dentures first and then an RPD with metal framework and clasps.The third group included 51 patients (disregarding the algorithm and lacking interim acrylic dentures) who refused interim acrylic dentures and received an RPD with metal framework and special elements.

### 2.3. OHIP5 Instrument

The shortest OHIP has 5 items. The OHIP-5 was devised to obtain information equivalent to 90% of the OHIP-49 summary score (with fewer questions than the OHIP-49) and does not categorize items into a set of seven domains. Five items are included in the OHIP-5: one for each of the four aspects of oral-health-related quality of life (OHRQoL)—oral function, orofacial pain, orofacial appearance, and psychosocial impact—and an additional item for oral function. The OHIP-5 has four dimension scores and one summary score [17,18,19].

The OHIP-5 questions were as follows:
Have you had difficulty chewing any foods because of problems with your teeth, mouth, dentures, or jaw?Have you had painful aching in your mouth?Have you felt uncomfortable about the appearance of your teeth, mouth, dentures, or jaws?Have you felt that there has been less flavor in your food because of problems with your teeth, mouth, dentures, or jaws?Have you had difficulty doing your usual jobs because of problems with your teeth, mouth, dentures, or jaws?

Answers were recorded on a 5-level Likert scale, with the authors indicating a coding from 0 to 4 (4—very often, 3—quite often, 2—occasionally, 1—almost never, and 0—never). The total score varies depending on the number of questions. The higher the score, the greater the impact of oral health issues on quality of life.

To answer each OHIP question, participants were questioned about the frequency with which they had encountered the problem in the preceding month.

Statistical Package for Social Sciences (SPSS, version 20) was used for data entry and analysis. We used nonparametric statistical tests to determine whether there were statistically significant differences between groups for each question. The Kruskal–Wallis test is the nonparametric equivalent of the Anova test. It shows whether there are statistically significant differences between the three groups for the participants’ answers to the five questions. However, to see which groups had statistically significant differences, we applied the Mann–Whitney test, the nonparametric equivalent of the t test for independent samples. We considered the null hypothesis to be that there were no statistically significant differences between the three groups; the research hypothesis was that there were statistically significant differences between the three groups. The statistical significance was set at *p* < 0.05.

## 3. Results

The final study group consisted of 338 participants, with an average age of 64.09 ± 7.38 (min. 49, max. 82). In total, 196 (58%) participants were female, and 70% of participants were from urban areas. Of all participants, 49.4% were employed, 36.4% were retired, and 14.2% were unemployed (Table 1).

Regarding the distribution of the participants according to the type of edentation, the statistical analysis showed that 173 (51.2%) of the subjects presented Kennedy class II edentation, followed by those with Kennedy class I (42.9%) and class III (5.9%) (Table 1).

Statistical analysis of questionnaire answers showed that subjects from groups 2 and 3 had a higher quality of life than subjects from group 1 (control) as evidenced by the increased frequency of their answers “never” (26.4% and 29.4%, respectively) and “almost never” (46.2% and 29.4%, respectively) recorded both in terms of difficulties chewing any foods and the aesthetic aspect of the smile.

However, there was no clear pattern when asked to rate how much they were affected by mouth discomfort, how much their meals tasted different, or how much trouble they had with their regular activities. The answer variants “almost never” and “occasionally” were more frequently selected for questions pertaining to the presence of oral cavity pain and the loss of taste in food. The answer variant “never” was more frequently selected for questions pertaining to the presence of difficulty in performing routine tasks, with the highest frequency recorded for group 1 (control), followed by group 3 (70%), and group 2 (53%). (Table 2).

The test results, shown in the Table 3, were expressed by a chi-square value with two degrees of freedom and were statistically significant (*p* = 0.0001 < 0.05) for questions 1, 2, 3, and 5. In these cases, the null hypothesis can be rejected, as there were statistically significant differences between the three groups. We used the Mann–Whitney test to determine whether there were significant differences between the various groups.

For question 1, scores for groups “1” and “2” differed significantly, with *p* = 0.0001 < 0.05; scores for groups “2” and “3” differed significantly, with *p* = 0.0001 < 0.05; however, scores for groups “1” and “3” did not differ significantly, with *p* = 0.615 > 0.05.

For question 2, scores for groups “1” and “2” differed significantly, with *p* = 0.0001 < 0.05; scores for groups ”2” and “3” differed significantly, with *p* = 0.0001 < 0.05; however, scores for groups “1” and “3” did not differ significantly, with *p* = 0.934 > 0.05.

For question 3, scores for groups “1” and “2” differed significantly, with *p* = 0.0001 < 0.05; scores for groups “1” and “3” differed significantly, with *p* = 0.001 < 0.05; however, scores for groups “2” and “3” did not differ significantly, with *p* = 0.925 > 0.05.

For question 4, scores for groups “1” and “2” did not differ significantly, with *p* = 0.732 > 0.05; scores for groups “1” and “3” did not differ significantly, with *p* = 0.548 > 0.05; and scores for groups “2” and “3” did not differ significantly, with *p* = 0.409 > 0.05.

For question 5, scores for groups “1” and “2” differed significantly, with *p* = 0.001 < 0.05; scores for groups “1” and “3” differed significantly, with *p* = 0.004 < 0.05; however, scores for groups “2” and “3” did not differ significantly, with *p* = 0.069 > 0.05.

## 4. Discussion

There is a substantial need to develop the materials and technology involved with RPDs because of the secondary expenses associated with the oral and systemic health implications of their usage. It is considered a reasonable and practicable therapeutic technique to use an RPD if it can replace lost structures while causing little damage to remaining hard and soft tissues [19].

A denture framework must be designed to ensure that the denture is strong and durable enough not to distort a patient’s repaired occlusion. An RPD’s success depends on a thorough understanding of RPD design and associated information, as well as effective communication with laboratory staff.

Various studies have shown that people who have lost teeth and need prosthodontic therapy have poor dental prosthetic status [20,21,22,23]. It is not uncommon for this group of patients to complain about aspects that are not actually incorrect [22,23]. Satisfying the needs of the patient should be the goal of any prosthodontic therapy [24]. The variables that characterize patients’ perceptions are different from those used by clinicians to evaluate clinical results. After the insertion of a prosthesis, patient satisfaction is determined by the patient’s level of physical health, psychological adjustment, social functioning, and the cost effectiveness of the therapy. A patient’s subjective opinion of the requirement for prosthodontic appliances can be compared to an examiner’s assessment of that need using the (WHO) diagnostic criteria [21,25] or the Geriatric Oral Health Assessment Index (GOHAI) [26]. 

Even though RPDs are often used to replace missing teeth, several issues with their use have been recorded in various populations [27,28,29,30].

In contrast to the findings of the current study, in several other studies, acrylic resin RPDs are significantly more prevalent than cast metal framework RPDs. The treatment approach that is ultimately chosen appears to be influenced by several factors, including the skills and knowledge of the dental laboratory technicians, and the intraoral circumstances [20,21,22].

Life expectancy is steadily rising in most countries throughout the world, including those that are still developing. It is anticipated that by 2030, approximately one billion individuals will be 65 years old or older, making up 13% of the population. The findings of the current study are helpful in assessing the patients’ quality of life when using RPDs, and such findings should be presented to patients. In addition, these findings offer valuable information that may be utilized for instructional and educational reasons [31,32]. 

To the best of authors’ knowledge, no clinical study has assessed the quality of life related to prosthetic treatment for populations in certain regions of Romania. Sex, educational background, dental attendance patterns, teeth brushing frequency, scarring experiences from childhood dentistry, the expense of dental care, and the care organization utilized were all connected with prosthodontic status differences, but not using a questionnaire regarding oral health [33,34]. 

Mc Entee et al. showed that approximately two-thirds of the older population suffers from poor dental health; yet, only one-third of those people reported having an issue with it. Approximately half of the individuals (54%) had difficulty with their dentures, and 83% did not have dentures, according to the researchers [35].

OHIP-measured denture status was revealed to be a substantial predictor of poor OHRQoL [16,36,37]. Frank et al. examined how satisfied patients were after an RPD was inserted. The most common sources of discontent regarding the new dentures were fit, ease of eating and chewing, hygiene of the mouth, speech, and sanitation [12]. In addition, according to the findings of Redford et al., the most common issues involving adaptation to denture use among new denture wearers were food becoming caught in the dentures, difficulty cleaning the dentures, discomfort or pain, poor retention, and how the dentures looked [38].

If RPD problems were related to denture factors or patient variables, the research could not determine this. The discontent of many patients cannot be explained simply by the poor quality of dentures.

For this reason, Romania has already implemented new programs and courses at the Faculty of Dental Medicine (Gerontostomatology) for dentists as well as oral hygienists. Dental personnel should be educated and re-educated regarding the physical, psychological, and social requirements of the elderly population, to provide better service. This study had certain limitations, including a small sample size and the fact that research participants cannot be assumed to represent the entire RPD-wearing population.

To strengthen the external validity of the epidemiological research regarding RPD acceptability and professional–patient appraisal of long-term functioning, more individuals from varied contexts should be included. Additional longitudinal research is necessary; ideally, it should include participants from a variety of regions and a larger sample size so that results will more accurately represent the people who live in this region of Romania.

## 5. Conclusions

Information regarding patients’ problems with RPDs and contributing variables will assist doctors in making educated decisions in the treatment of partly edentulous patients requiring RPDs, as well as in reducing possible resource waste.

Although this study was confined to a specific group, it can provide insight into RPD patients’ happiness when treated in an academic environment. If a student-treated patient sample does not apply to other government institutions or a commercial dental practice because of differences in patient populations, quality control, and treatment planning criteria, this should be highlighted.

A combination of research and clinical exams is needed to determine the impact of many factors on patient satisfaction with dentures, such as the status of abutment teeth, denture-bearing regions, oral mucosal health, saliva quality, and oral hygiene habits. This research, despite its limitations, provides a general view of RPD wearers’ satisfaction in the N-E region of Romania.

## Figures and Tables

**Table 1 medicina-58-01247-t001:** Demographic characteristics of the study group.

		No	%
Age	64.09 ± 7.38 (min. 49, max. 82)		
Sex	Female	196	58.0
	Male	142	42.0
Residence	Urban	241	71.3
	Rural	97	28.7
Occupation	Employee	167	49.4
	Unemployed	48	14.2
	Retired	123	36.4
Type of edentation	Class I Kennedy	145	42.9
	Class II Kennedy	173	51.2
	Class III Kennedy	20	5.9

**Table 2 medicina-58-01247-t002:** Comparison of satisfaction between the three groups of RPD wearers.

OHIP 5-ITEM		LOT 1	LOT 2	LOT 1	LOT 3	LOT 2	LOT3
Have you had difficulty chewing any foods because of problems with your teeth, mouth, dentures or jaw?	Mean rank	90.54	175.31	77.80	81.49	129.85	69.14
	Mann–Whitney U	326.5		2576		2200	
	*p*	0.000		0.615		0.000	
Have you had painful aching in your mouth?	Mean rank	109.37	164.28	116.32	117.15	68.44	100.95
	Mann–Whitney U	5922.5		4582.5		1583.5	
	*p*	0.000		0.934		0.000	
Have you felt uncomfortable about the appearance of your teeth, mouth, dentures or jaws?	Mean rank	115.48	160.7	124.07	89.64	79.22	78.55
	Mann–Whitney U	6570		3245.5		2680	
	*p*	0.000		0.001		0.925	
Have you felt that there has been less flavor in your food because of problems with your teeth, mouth, dentures or jaws?	Mean rank	146.12	142.76	117.86	111.67	81.01	74.82
	Mann–Whitney U	9368.5		4369		2490	
	*p*	0.732		0.548		0.409	
Have you had difficulty doing your usual jobs because of problems with your teeth, mouth, dentures or jaws?	Mean rank	112.39	162.51	122.57	94.94	75.95	85.34
	Mann–Whitney U	6242		3516		2379.5	
	*p*	0.000		0.004		0.069	

Mann–Whitney test, *p* < 0.05.

**Table 3 medicina-58-01247-t003:** Patients’ satisfaction with RPD usage.

OHIP 5-ITEM	Lot	N	Variables					Mean Rank	Chi -Square	df	*p*
			Never	Almost Never	Occasionally	Quite Often	Very Often				
Have you had difficulty chewing any foods because of problems with your teeth, mouth, dentures or jaw?	LOT1	106	4.4%	13.8%	39.2%	26.0%	16.6%	114.84	86.542	2	0.000
	LOT2	181	26.4%	46.2%	16.0%	8.5%	2.8%	214.15			
	LOT3	51	29.4%	29.4%	27.5%	9.8%	3.9%	124.63			
Have you had painful aching in your mouth?	LOT1	106	17.0%	55.7%	18.9%	6.6%	1.9%	124.31	37.180	2	0.000
	LOT2	181	3.9%	35.9%	37.6%	13.3%	9.4%	189.60			
	LOT3	51	2.0%	33.3%	47.1%	11.8%	5.9%	192.10			
Have you felt uncomfortable about the appearance of your teeth, mouth, dentures or jaws?	LOT1	106	48.1%	34.9%	10.4%	4.7%	1.9%	141.20	26.716	2	0.000
	LOT2	181	21.5%	42.0%	26.5%	6.1%	3.9%	193.77			
	LOT3	51	52.9%	25.5%	15.7%	5.9%	0.0%	142.19			
Have you felt that there has been less flavor in your food because of problems with your teeth, mouth, dentures or jaws?	LOT1	106	7.5%	24.5%	33.0%	25.5%	9.4%	173.63	0.664	2	0.718
	LOT2	181	10.5%	25.4%	32.0%	22.7%	9.4%	169.63			
	LOT3	51	9.8%	33.3%	23.5%	23.5%	9.8%	160.49			
Have you had difficulty doing your usual jobs because of problems with your teeth, mouth, dentures or jaws?	LOT1	106	85.8%	11.3%	1.9%	0.9%	0.0%	134.49	37.326	2	0.000
	LOT2	181	53.0%	24.9%	5.5%	9.9%	6.6%	194.09			
	LOT3	51	70.6%	15.7%	7.8%	5.9%	0.0%	154.28			

## Data Availability

Data that support the findings of this study are available on request from the corresponding author.

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
