# Peer review of "An Observational Study on Oral Health and Quality of Life for RPD Wearers in the N-E Region of Romania"

_medicina, 2022, doi:10.3390/medicina58091247_

Round 1
Reviewer 1 Report
The manuscript sounds interesting, and the work is well written and well planned. References should be written according to the auther's guide. https://www.mdpi.com/journal/medicina/instructions#references Comma at end of sentence: "." is missing in a number of cases.Author Response
Dear Reviewer ,
I would like to thank You once again for precisely done review. It was a great experience to follow all the indications and comments that You gave us. We followed all Your recommendations which increased the value of this manuscript, and gave us additional, valuable knowledge.
Best regards,
Yours sincerely,
on behalf of authors
Balcoș Carina

Reviewer 2 Report
This paper included a considerable number of participants and the research question is interesting. However, many methodological aspects should be improved.
The authors did not provide any indication of the study design, which should be obvious from the title of the study to begin with. It is obviously an observational study and STROBE checklist should be followed (https://www.strobe-statement.org/). They also use abbreviations without giving the full name in the first place (RPD).
The Material and Methods and Results sections should be improved according to methodological requirements.
The main problem of the study design is that the control group is missing and the demographic data of the included participants are not provided.
The statistical methods should be improved as well as the presentation of the results.
Tables of results should be understandable without the rest of the manuscript. They must include information about all abbreviations and statistical tests (the legend).
Author Response
Balcoș Carina, Ph.D., DMD.
Lecturer
Department Of Surgery,
Faculty of Dentistry,
University of Medicine and Pharmacy Grigore T.Popa
Iasi, Romania
email: carina.balcos@umfiasi.ro
23.08.2022
Dear Reviewer,
I would like to thank You once again for precisely done review. It was a great experience to follow all the indications and comments that You gave us. We followed all Your recommendations which increased the value of this manuscript, and gave us additional, valuable knowledge.
Best regards,
Yours sincerely,
on behalf of authors
Balcoș Carina
Response to Reviewer 2 Comments
The article " An observational study on oral health and quality of life for RPD wearers in the N-E region of Romania”
Point 1. The authors did not provide any indication of the study design, which should be obvious from the title of the study to begin with. It is obviously an observational study and STROBE checklist should be followed (https://www.strobe-statement.org/). They also use abbreviations without giving the full name in the first place (RPD).
Response 1: We tried to improve the initial form of the manuscript concerning all the sections, following the steps provided by Strobe checklist including the title of the paper and we explained the abbreviations in the first place.
Point 2. The Material and Methods and Results sections should be improved according to methodological requirements.
Response 2: At the material and method section we tried to follow the proper procedure form the same Strobe checklist and added demographic information about the study group in results as can be seen in the paper.
Point 3. The main problem of the study design is that the control group is missing and the demographic data of the included participants are not provided
Response 3: We tried to make clearer the study procedures so we inserted a control group, the one with patients with good stability and retention for dentures and compliance for clinic's protocol.
Point 4. The statistical methods should be improved as well as the presentation of the results.
Response 4: We tried to improve the tables with statistical information related to the distribution of the answers in the form of percentages to the 5 questions in the questionnaire. Also, we introduced the table with the demographic characteristics of the study group.
Point 5. Tables of results should be understandable without the rest of the manuscript. They must include information about all abbreviations and statistical tests (the legend).
Response 5:We tried to make all the necessary changes concerning the abbreviations and statistical test.
